# The Anticonvulsant Effects of Different Cannabis Extracts in a Zebrafish Model of Epilepsy

**DOI:** 10.3390/biom15050654

**Published:** 2025-05-01

**Authors:** Karen Jackson, Maytal Shabat-Simon, Jonathan Bar-On, Rafi Steckler, Soliman Khatib, Snait Tamir, Paula Adriana Pitashny

**Affiliations:** 1Faculty of Sciences and Technology, Tel-Hai College, Tel-Hai 1220800, Israel; karen@migal.org.il (K.J.); maytalsh@telhai.ac.il (M.S.-S.); ybaron36@gmail.com (J.B.-O.); rafi@telhai.ac.il (R.S.); solimankh@migal.org.il (S.K.); snait@telhai.ac.il (S.T.); 2MIGAL Galilee Research Institute, Kiryat Shmona 1101600, Israel

**Keywords:** epilepsy, zebrafish, cannabis extracts, CBD, entourage effect

## Abstract

Epilepsy is a widespread neurological disorder that remains a critical global public health challenge. While numerous antiepileptic drugs (AEDs) are available, many patients either fail to achieve adequate seizure control or experience significant side effects. One promising alternative is pure cannabidiol (CBD), but using a whole cannabis extract may be equally effective and preferred for some patients. In the current study, we employed the pentylenetetrazole (PTZ)-induced hyperactivity model in zebrafish to compare the effects of CBD with various cannabis extracts. We evaluated three cannabis strains, each subjected to three different extraction methods, and benchmarked the results against the commercially available AED valproic acid (VPA). Our findings revealed that 5.7 µg/mL of CBD and 10 µg/mL of different extracts significantly reduced movement compared to PTZ and VPA. In addition, effective extracts produced effects similar to pure CBD despite containing much lower molecule levels. These results reinforced and expanded previous evidence supporting the clinical potential of both CBD and whole cannabis extracts for seizure control while suggesting a possible entourage effect. Further research is necessary to determine which patients may benefit more from pure CBD versus those who might prefer whole cannabis extracts.

## 1. Introduction

Epilepsy is one of the most prevalent neurological disorders, affecting approximately 50 million people worldwide, including up to 1% of children [1,2]. This disorder is marked by recurrent, unprovoked seizures that elevate the risk of injury; premature mortality; and diminished participation in daily activities, education, and employment [3]. The seizures result from excessive and synchronized electrical discharges in the brain [4], yet can occur even when no detectable brain abnormalities are present [5,6]. They manifest in diverse forms, ranging from brief lapses in consciousness or subtle involuntary movements to violent full-body convulsions, underscoring this disorder’s varied nature [7]. In addition to the various forms of epilepsy, several other diseases may also present with epileptic seizures as a symptom [8].

Addressing the burden of seizures remains a global public health priority [2], although there are several antiepileptic drugs (AEDs) that target distinct mechanisms involved in seizure activity. Older AEDs, such as Carbamazepine, block sodium channels to reduce neuronal excitability, while benzodiazepines, such as Clobazam, are positive allosteric modulators of synaptic GABA_A_ receptors [9,10,11]. Valproic acid (VPA), a commonly used AED, not only blocks sodium channels but also enhances GABA levels and reduces calcium influx, making it effective against various types of seizures, including generalized and absence seizures [12,13,14,15]. Yet, despite advancements in treatment, approximately 30% of patients do not achieve adequate seizure control with available AEDs, leading to refractory epilepsy [16,17]. Moreover, AEDs are associated with side effects that may lead to treatment discontinuation or deteriorated quality of life. Common side effects include headaches, nausea, and dizziness, while more adverse effects include auditory and visual problems, skin problems, liver dysfunction, pancreatitis, and kidney disorders. Some AEDs may even result in life-threatening conditions as well as severe abnormalities, especially in patients with comorbidities or in pregnant women [18]. For example, VPA has been associated with disruptions in normal brain development, potentially leading to cognitive and behavioral deficits; an increased risk of neurodevelopmental disorders and impairments; and alterations in neurogenesis, gliogenesis, and synaptic function [19,20,21,22]. Hence, the search for safe and effective novel treatments continues.

Identifying new AEDs for treating epilepsy and seizures has mostly been carried out using rodent models [23]. However, using zebrafish (*Danio rerio*) may allow more rapid scanning of novel therapeutics. Zebrafish offer practical benefits, including their high productivity, rapid development, and small size, which make them ideal for high-throughput, cost-effective drug screening [24]. Their larvae allow even higher throughput behavioral screening compared to adult fish [25,26,27]. In both rodent and zebrafish models, epilepsy can be induced using a convulsant like pentylenetetrazole (PTZ), with rodents typically presenting limb clonus, rearing and falling, and tonic–clonic convulsions [28,29] and zebrafish typically presenting distinct changes in swimming patterns, such as increased hyperactivity [30,31,32,33,34,35,36]. This increase in movement serves as a biomarker of seizure activity and correlates strongly with electrographic discharges recorded from the zebrafish brain, mirroring the relationship between EEG activity and behavioral manifestations in rodent models [27]. The use of distance traveled as a measure of seizure propensity is further justified by its ability to detect dose-dependent responses to convulsants and anticonvulsants. Indeed, a systematic review of chemically induced seizure models in zebrafish found distance traveled to be among the most commonly assessed behavioral outcomes, serving as a primary endpoint in nearly 50% of the 178 reviewed studies [37]. Zebrafish models have also emerged as valuable for AED research due to the genetic, physiological, and neurochemical similarities between zebrafish and humans, particularly within the central nervous system [38]. Key areas in the brains of zebrafish after 5 days postfertilization (dpf) are homologous to those observed in mammals and hence allow comparisons between the patterns of zebrafish behavior in different laboratory tests and the activity of older mice and rats [39].

Recent examples of the use of zebrafish to examine anticonvulsant effects include studies reporting that phytocannabinoids, including cannabidiol (CBD), can effectively reduce PTZ-induced hyperactivity in these models [40,41,42]. CBD is a non-psychoactive compound derived from *Cannabis sativa* that has demonstrated promising anticonvulsant properties in various animal seizure models, with potential mechanisms involving the endocannabinoid system and calcium signaling pathways [43,44]. Notably, the endocannabinoid system is highly conserved between zebrafish and mammals, including humans, with high densities of CB1 receptors in areas that bear signs of functional homology to the mammalian hippocampus [45], which is the seizure-initiating zone in many epilepsy patients as well as in animal models of epilepsy [39].

CBD has gained significant attention for its anticonvulsant properties, often being studied in isolation to delineate its specific mechanisms of action. Moreover, the FDA approved a CBD-based AED (Epidiolex) based on evidence from clinical trials involving patients with Lennox–Gastaut or Dravet syndrome [46,47]. However, the therapeutic potential of whole cannabis extracts, which contain a full spectrum of cannabinoids, terpenes, and other bioactive compounds such as Δ^9^-tetrahydrocannabinol (Δ^9^-THC), is increasingly recognized. Several of these compounds demonstrated therapeutic potential due to anxiolytic, antidepressant, antitumor, immunomodulatory, and neuroprotective effects [48,49,50,51,52,53]. Moreover, these compounds may act synergistically with CBD, enhancing its effects through what is known as the “entourage effect” [48,54,55]. This phenomenon suggests that a combination of multiple cannabis-derived compounds may provide a broader or more potent therapeutic benefit compared to CBD alone. For this reason, many individuals and healthcare practitioners lean toward the use of whole cannabis extracts, valuing their perceived efficacy and the holistic approach they offer in managing complex conditions [56,57,58].

Here, we aimed to use a zebrafish epilepsy model where generalized seizures are induced by PTZ to investigate the anticonvulsant effects of whole cannabis extracts from three distinct strains. Each strain was prepared through three different extraction methods, yielding varying compositions. The therapeutic potential of the extracts was then compared to that of a positive control (VPA) and that of pure CBD at different concentrations.

## 2. Materials and Methods

### 2.1. Zebrafish Husbandry

Zebrafish (*Danio rerio*) were maintained using standard animal care protocols [59]. Wildtype adult AB/zebrafish were housed in a recirculating aquatic system (Aquazone, Tzofit, Israel) set to 28.5 ± 1 °C with a pH range of 7.0–7.2 and a 14:10 h light–dark cycle. Embryos from multiple breeding pairs were collected, pooled in Petri dishes containing E3 medium (5 mM NaCl, 0.17 mM KCl, 0.33 mM CaCl_2_-2H_2_O, 0.33 mM MgSO_4_-7H_2_O, and 10 mM HEPES (pH 7.2)), and examined to remove unfertilized ones. The fertilized embryos were then placed in an incubator maintained at 28 °C.

### 2.2. Pure CBD

Pure CBD (obtained from Sigma-Aldrich, Rehovot, Israel #C7515) was diluted in E3 medium, maintaining a final DMSO (dimethyl sulfoxide, Sigma-Aldrich, Rehovot, Israel #472301-1L) concentration of 0.1% *w*/*v*, to obtain concentrations of 6, 9, 12, 15, and 18 µM (equivalent to 1.9, 2.8, 3.8, 4.7, and 5.7 µg/mL, respectively).

### 2.3. Cannabis Extractions

Inflorescences from various cannabis strains were obtained from a cannabis farm (Brlev Agricultural Crops Ltd., Ein Vered, Israel). Three distinct extractions were performed on the inflorescences from each strain. Equal amounts of the inflorescences (330 mg per strain) were distributed into three 15 mL test tubes. Two tubes were incubated at 25 °C, while the third was incubated at 100 °C for 1 h to decarboxylate cannabinoid acids. Next, 10 mL of ethyl acetate (EA) was added to the first tube, while 10 mL of ethanol was added to the second (E25) and third (E100) tubes. Each solution was mixed using a magnetic stirrer for 90 min. The mixtures were centrifuged at 10,000 rpm for 10 min, after which the upper layers were collected. The solid phase was re-extracted twice, using 3 mL of solvent each time. The upper layers from all three extractions were combined and evaporated under nitrogen. The resulting dried extracts were stored at −80 °C until use.

The dried cannabis extracts were solubilized in DMSO at 200 mg/mL and stored at −20 °C. Immediately before the seizure assays, the stored solutions of the cannabis extracts were diluted in E3 medium to reach the desired concentrations, maintaining a final DMSO concentration of 0.1% *w*/*v*. E3 medium with 0.1% DMSO served as the vehicle control, while E3 medium alone acted as the negative control.

### 2.4. Seizure Assay

The experimental procedure is presented in Figure 1. At 5 dpf, zebrafish larvae were individually placed into a 96-well plate, each equipped with a 100-mesh filter insert (MultiScreen-MESH Filter Plate, Darmstadt, Germany) containing 350 μL of buffered E3 medium, and allowed to acclimate at 28 °C for 24 h. On the test day, the plate was transferred to a DanioVision™ observation chamber, a system designed for high-throughput tracking of zebrafish larvae and other small organisms, and left for an additional 30 min of acclimation. Subsequently, the insert plate containing the larvae was moved to a treatment plate containing either the negative control (0.1% DMSO), the positive control (5 mM VPA; Merck, Herzliya, Israel #PHR1061), or the test substances (CBD or the cannabis extracts at varying concentrations), and movement was recorded for 30 min. Following the 30 min exposure, a viability test was conducted visually and by placing the larvae in darkness for 3 min in E3 medium to observe their movement patterns. Larvae that completely lost mobility were excluded from the experiment. The insert was then transferred to a solution containing 15 mM PTZ (Sigma-Aldrich, Rehovot, Israel #6500-25G) for 30 min, during which motor activity was recorded to assess the anticonvulsant potential of the test compounds.

### 2.5. Video Tracking and Analysis

Behavioral testing was conducted using a DanioVision™ (Noldus Information Technology, Wageningen, The Netherlands, Israeli provider: New Bio Technology Ltd. (NBT), Or Akiva, Israel) lightproof recording chamber equipped with an infrared camera. The activity was measured with EthoVision XT13 (Version number: 13, Noldus) software, which recorded the total distance traveled, segmented into 1 min intervals over 30 min. Each experimental group consisted of 12 larvae per treatment for CBD and the different extracts, with a minimum of three replicates per CBD or extract concentration. As there were no significant differences between the groups, larval activity was pooled for each treatment group (for a total of n = 36/group), while all controls were pooled into a single group (for a total of n = 108/group) for further analysis.

### 2.6. Characterizing Extracts’ Chemical Compositions

The cannabinoids in the various cannabis extracts were isolated, identified, and quantified using ultra-high-performance liquid chromatography coupled with mass spectrometry (UHPLC-MS; Ultimate 3000, Thermo Fisher Scientific, Airport City, Israel) and detected by a Q Exactive™ Plus Hybrid Quadrupole-Orbitrap (Thermo Fisher Scientific, Airport City, Israel). The sample injection volume was 5 μL. A standard solution containing 14 cannabinoid standards purchased from Merck (Herzliya, Israel) was injected under identical conditions for comparison. Separation was achieved using an EC-C18 column (150 mm length, 3.0 mm diameter, and 2.7 μm particle size; Agilent Technologies, Tel Aviv, Israel). The mobile phase consisted of 0.1% formic acid in double-distilled water (DDW) and 0.1% formic acid in acetonitrile (ACN), with a 0.4 mL/min flow rate. Gradient elution was performed as follows: 0–3 min, 60% ACN; 3–7 min, 60–80% ACN; 7–12 min, 80–90% ACN; 12–20 min, 90% ACN; 20–22 min, 90–60% ACN; and 22–25 min, 60% ACN. The column was maintained at 30 °C, and the autosampler was maintained at 10 °C. Ionization was performed in both positive and negative modes. The instrument parameters included a capillary voltage of 3500 V; a capillary temperature of 350 °C; and nitrogen gas flow rates of 35 mL/min for sheath gas, 10 mL/min for auxiliary gas, and 1 mL/min for sweep gas. The mass range was set from 150 to 800 m/z with a resolution of 70,000. Data analysis was conducted using Xcalibur and Freestyle software (Thermo Fisher Scientific, Airport City, Israel, Version number: 4.2).

The terpenes from each extract were isolated and analyzed using gas chromatography–mass spectrometry (GC-MS; Thermo Fisher Scientific, Airport City, Israel TSQ 8000). Analysis was performed on an Equity-1 capillary column (60 m length, 0.25 mm internal diameter, and 0.25 μm particle size). The injector temperature was set to 280 °C, with a sample injection volume of 1 μL. Helium gas was used as the mobile phase at a 1 mL/min flow rate. The column temperature was programmed to increase by 5 °C/min from 60 °C to 275 °C. The terpenes in the extracts were identified and quantified by comparing their retention times to those of a standard solution (containing 34 abundant cannabis terpenes purchased from Merck, Herzliya, Israel) injected in the GC-MS system. Peaks with retention times differing from the standard solution were further analyzed by comparison with the NIST library.

### 2.7. Statistical Analysis

The data were analyzed using a one-way ANOVA with a 95% confidence interval. If no significant differences were found between the replicates, the larval activity data were pooled into treatment and control groups for further analysis. The data were analyzed in SPSS (Version number: 28.0.1.1(14)). The distances moved were compared using a one-way ANOVA, and significant effects were further examined using the Tukey post hoc test. The data are presented as means ± SEM. In the first phase, CBD at different concentrations was compared to the negative (DMSO) and positive (VPA) controls. In the second phase, this was performed for the extracts obtained from the different strains using different extraction methods and concentrations. Finally, the effective treatments from the two phases were compared to identify favorable treatment options.

## 3. Results

### 3.1. Behavioral Effects

#### 3.1.1. Lack of Sedative Effects

During the treatment phase, the comparable distances moved between the larvae exposed to the extracts and those exposed to the negative control indicated that the tested compounds did not cause sedation (Appendix A). Furthermore, we conducted a viability test, which revealed an increase in movement following exposure to darkness (3 min), suggesting that the cannabis extracts did not affect the behavioral reaction to darkness (Appendix A). An exception was observed in one group (Appendix A; CAN 2–E100—10 µg/mL), which did not show an increase in movement following darkness exposure. However, the behavioral results in Appendix A did not reveal any abnormal total distance moved values for this treatment (E100—10 µg/mL; CAN 1, 2, and 3). None of the larvae were excluded from the experiments after the viability test.

#### 3.1.2. CBD Reduces PTZ-Induced Hyperactivity

An ANOVA revealed a main group effect (F_7,376_ = 79.14, *p* < 0.001), and post hoc analysis revealed significantly reduced distance moved values compared to PTZ following exposure to VPA and CBD at concentrations of 1.9, 2.8, 3.8, 4.7, and 5.7 µg/mL (Figure 2). Further analysis revealed that, compared to VPA, there were significantly greater distance moved values following exposure to 1.9 and 2.8 µg/mL CBD and a significantly reduced distance moved value following exposure to 5.7 µg/mL CBD. The 3.8, 4.7, and 5.7 µg/mL CBD concentrations were selected for subsequent analyses. Full descriptive statistics, effect sizes, and post hoc results are presented in Appendix A.

#### 3.1.3. Extracts Reduce PTZ-Induced Hyperactivity

An ANOVA revealed a main group effect for all strains (CAN-1: F_14,741_ = 63.752, *p* < 0.001; CAN-2: F_14,741_ = 55.628, *p* < 0.001; CAN-3: F_14,741_ = 59.871, *p* < 0.001), and post hoc analysis revealed significantly reduced distance moved values following exposure to 10 µg/mL of the different extracts (Figure 3). At that concentration, across strains, movement was reduced for all extracts compared to PTZ, and compared to VPA, it was reduced for the E25 and E100 methods. For all other concentrations, there were no differences when compared to PTZ, while the distance moved was significantly higher compared to VPA. Complete descriptive statistics, effect sizes, and post hoc results are presented in Appendix A.

Hence, the concentration of 10 µg/mL was used for all strains and extraction methods in the following analyses.

#### 3.1.4. Comparing Effective CBD and Extract Concentrations

For CAN-1 (Figure 4A), an ANOVA revealed a main group effect for the cannabis extract and CBD concentrations (F_5,246_ = 8.63, *p* < 0.001). Post hoc analysis revealed that the distance moved values were significantly reduced for the E25 extract and 5.7 µg/mL CBD compared to 3.8 µg/mL CBD and the EA extract. For CAN-2 (Figure 4B), an ANOVA revealed a main group effect for the cannabis extract and CBD concentrations (F_5,246_ = 15.27, *p* < 0.001). Post hoc analysis revealed that the distance moved values were significantly reduced for the E100 extract and 5.7 µg/mL CBD compared to E25, EA, and 3.8 µg/mL CBD. Additional results included increased distance moved for the EA extract compared to 4.7 µg/mL CBD and reduced distance moved for 5.7 µg/mL CBD compared to 4.7 µg/mL CBD. For CAN-3 (Figure 4C), an ANOVA revealed a main group effect for the cannabis extract and CBD concentrations (F_5,246_ = 20.24, *p* < 0.001). Post hoc analysis revealed that, similar to CAN-2, the distance moved values were significantly reduced for the E100 extract and 5.7 µg/mL CBD compared to 3.8 µg/mL CBD, E25, and EA. E100 from CAN-3 reduced PTZ-induced hyperactivity slightly better than E100 from CAN-2. This was observed despite the viability test results, which showed that E100 from CAN-3 led to an increase in movement following darkness exposure, whereas E100 from CAN-2 did not (Appendix A), suggesting that E100 from CAN-2 exerted other behavioral effects rather than sedation.

Additional results included increased distance moved for the EA extract compared to 3.8 and 4.7 µg/mL CBD and increased distance moved for the E25 extract compared to 4.7 µg/mL CBD. The effect sizes and post hoc results are presented in Appendix A.

### 3.2. Extract Profiles

The concentrations of the cannabinoids in the solutions of the different extracts at 10 µg/mL are presented in Table 1. CAN-1 included high levels of THC and THCA, with elevated levels of THC following heating. The CBD levels were considerably lower, with a minor increase following heating. CAN-2 included high levels of CBD and CBDA following all extraction methods. The THC levels were considerably lower but increased following heating. CAN-3 included relatively high levels of both THC and CBD, which further increased following heating but were still around half the levels in CAN-1 and CAN-2, respectively.

The concentrations of the terpenes in the 10 µg/mL solutions of the different extracts are presented in Table 2. Overall, the terpenes demonstrated a much more uniform distribution than the cannabinoids. For CAN-1, there were higher levels of Beta-Caryophyllene when the extract was not heated (0.0109 and 0.0113 µg/mL for the E25 and EA methods). For CAN-2, there were higher levels of Alpha Bisabolol for all methods (0.0151, 0.0126, and 0.014 µg/mL for the E25, E100, and EA methods), while CAN-3 lacked a distinguishing characteristic.

## 4. Discussion

In this work, we set out to explore for the first time the anticonvulsant effects of whole cannabis extracts on PTZ-induced hyperactive larval movement. We found that 5.7 µg/mL of CBD and 10 µg/mL of different extracts significantly reduced movement compared to PTZ and VPA. For the THC-rich CAN-1, the E25 method seemed to present the best effect (with only 0.015 µg/mL of CBD, which was 337-fold less concentrated than the best performance of pure CBD), while for the CBD-rich CAN-2 and the intermediate CAN-3, the E100 method seemed to present the best effects (with 5.174 µg/mL of CBD for CAN-2 and 2.661 µg/mL for CAN-3). E100 from CAN-3 demonstrated slightly better performance than the most effective pure CBD concentration, despite containing only half the CBD content. These latter findings align with the historically popular methods of cannabis consumption, which primarily involve heating in various forms, with inhalation through cannabis-based cigarettes being the most common approach over time [60] (heating decarboxylate THCA and CBDA into the active forms of CBD and THC [48]).

The results suggest that pure CBD and cannabis extract may be used for different applications or individuals. On the one hand, the fact that purified CBD can induce effects similar to those of the whole extract makes it a better candidate for clinical application due to the known concentration of the active ingredient, the lack of possible psychoactive effects from Δ^9^-THC, and the consequent reduced regulatory challenges. Moreover, studies have indicated that concomitant CBD and Clobazam appear to produce additive benefits and may represent a promising approach to control seizures while reducing the side effects linked to Clobazam [60]. Similarly, while larvae treated with VPA exhibited significantly reduced survival rates compared to controls, co-treatment with VPA and CBD resulted in survival rates comparable to those of untreated controls. Notably, the combined treatment largely mitigated the adverse effects typically observed when each compound was administered individually [61]. However, purified CBD was also associated with more frequent mild and severe adverse effects compared to CBD-rich extracts [62]. On the other hand, natural compounds, such as whole cannabis extracts, are widely used as complementary natural therapies to manage seizures and other conditions [63,64]. For example, for chronic neuropathic pain, the benefit–safety profiles of cannabinoids were found to be superior to other commonly used medications, mainly because they contribute more to quality of life and have a more favorable side effect profile [65]. Similarly, the majority of recreational and medical cannabis users report reducing or stopping their use of more conventional sleep aids after initiating cannabis use [66]. However, responses to cannabis are dose-dependent, and with vast individuality in responses, the same dose may benefit some but not others [67]. For example, while most children treated with a formula that contained CBD and Δ^9^-THC (20:1 in olive oil) reported a reduction in seizure frequency, a minority reported aggravation of seizures [68].

Our results are relevant to the current debate in the literature regarding the possibility of an entourage effect. On the one hand, there are indications that whole cannabis extract has advantages compared to single isolates [69]. For example, a meta-analysis found that CBD-rich extracts seem to present a better therapeutic profile than purified CBD in patients with refractory epilepsy [62]. On the other hand, recent reviews concluded that there is a lack of sound evidence supporting the existence of an entourage effect [70] and that additive enhancement of cannabinoid efficacy by terpenes remains unproven [54]. In zebrafish larvae, it was shown that combined exposure to CBD and Δ^9^-THC appears to have a synergistic effect on PTZ-induced behavior [42]. However, there was also a significant reduction in the activity of control larvae, suggesting that Δ^9^-THC produces a general sedation that partially explains the decrease in the PTZ-induced activity pattern. Here, we showed that the effective extracts produced effects similar to pure CBD while containing much lower concentrations of this molecule. In addition, beneficial effects were observed in strains that contained low levels of Δ^9^-THC. Hence, our results support an entourage effect, although it is unknown whether it was facilitated by cannabinoids that demonstrated anti-seizure effects in previous publications (e.g., [40]) or by combinations of cannabinoids and terpenes with supporting activity.

Our results suggest a beneficial effect of pure CBD at a higher concentration of 5.7 µg/mL (or 18 µM) compared to prior studies [40,41,42], which reported significant effects at concentrations of 0.6–4 µM. This may be explained by protocol differences, including variations in the exposure time, PTZ concentration, and monitoring conditions, which influence the effective CBD dose. These differences highlight the need to verify optimal CBD dosing within each study’s specific conditions before making comparisons. In addition, the purpose of using two different solvents, ethanol and ethyl acetate, for cannabis extraction was to compare the relative contributions of cannabinoids and terpenes. Ethyl acetate was expected to extract higher concentrations of less polar terpenes, but the results revealed few differences between the two solvents.

Despite the contributions of this study to our understanding of the anticonvulsant effects of cannabis extracts and CBD in a zebrafish model, several limitations warrant consideration. The use of the zebrafish model, while beneficial due to its genetic and physiological similarities to mammals, may not fully replicate the complexity of human epilepsy. As such, the translational relevance of these findings to human clinical practice should be carefully evaluated. In addition, we only examined distance traveled and therefore did not examine the full range of behaviors that zebrafish exhibit during PTZ-induced seizures. Distance traveled is a well-established and reliable parameter for assessing seizure activity in zebrafish larvae [26,37,41,42,71,72], and the use of automated tracking systems further ensures precise measurement of hyperlocomotion, reducing observer bias and enhancing reproducibility. This study also focused on a limited selection of cannabis strains and extraction methods, which may not encompass the full pharmacological diversity inherent in cannabis. The cannabis plant contains a wide array of cannabinoids and terpenes, and their interactions and contributions to anticonvulsant effects remain incompletely understood. Future studies incorporating a broader range of cannabis cultivars, extraction techniques, and specific cannabinoid profiles are essential for elucidating their potential therapeutic properties. In the same line, we only examined the effects of the extract prior to PTZ exposure. However, as many AEDs are rescue medications, future studies should examine whether CBD and the different extracts arrest seizures or reduce seizure severity when administered after application of PTZ. Additionally, this study primarily assessed short-term behavioral outcomes, limiting its ability to assess long-term efficacy and safety. Longitudinal investigations are necessary to determine the sustainability of the therapeutic effects and identify any delayed adverse effects. Lastly, the concentration range of the CBD and cannabis extracts examined in this study was relatively narrow and may not have captured the full spectrum of their dose-dependent effects. Further dose–response studies are required to define the optimal therapeutic window better and to identify thresholds for maximal efficacy and safety. Addressing these limitations in future research will provide a more comprehensive understanding of the potential clinical applications of cannabis-based treatments for epilepsy.

## 5. Conclusions

In conclusion, our results validate and extend previous ones concerning the potential clinical effects of both CBD and whole cannabis extracts used for seizure control. Further research exploring broader ranges of strains, concentrations, and extraction methods and including more extended observation periods is needed to establish the contributions of terpenes and minor cannabinoids to seizure control and to allow personalized medicine.

## Figures and Tables

**Figure 1 biomolecules-15-00654-f001:**
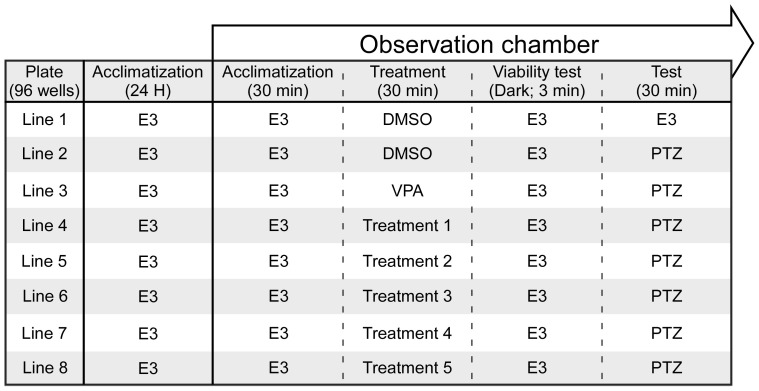
The experimental procedure. Following acclimatization, zebrafish larvae were exposed to the different solutions during the treatment phase, with DMSO and VPA serving as controls. Viability was then assessed visually and during exposure to darkness (movement was recorded). Finally, during the test phase, movement was recorded during exposure to PTZ.

**Figure 2 biomolecules-15-00654-f002:**
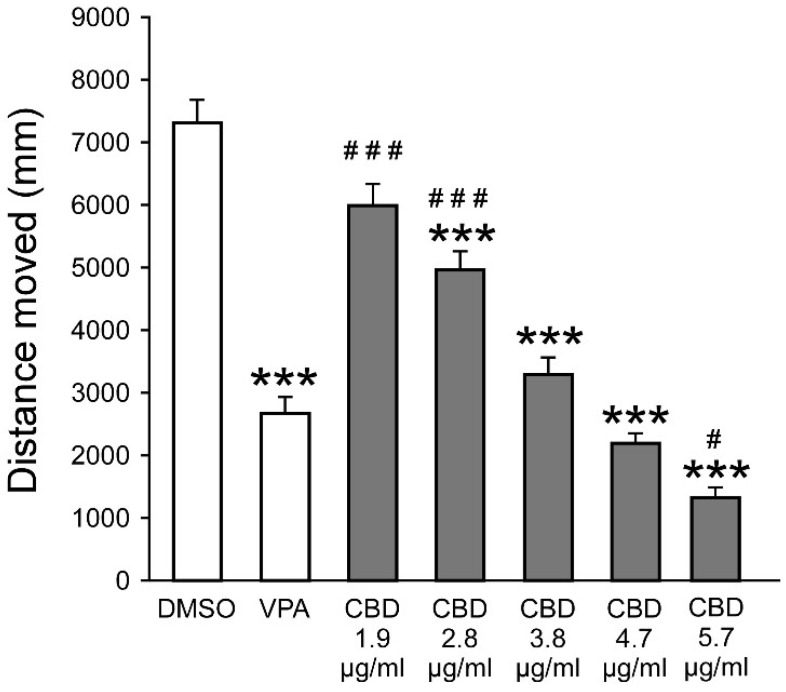
Effect of CBD concentration on PTZ-induced larval movement. Total distance moved (mm) by zebrafish larvae during 30 min of PTZ exposure following pre-treatment with DMSO, VPA (positive control), or varying concentrations of CBD. *** *p* < 0.001 vs. PTZ; ^#^ *p* < 0.05 vs. VPA; ^###^ *p* < 0.001 vs. VPA.

**Figure 3 biomolecules-15-00654-f003:**
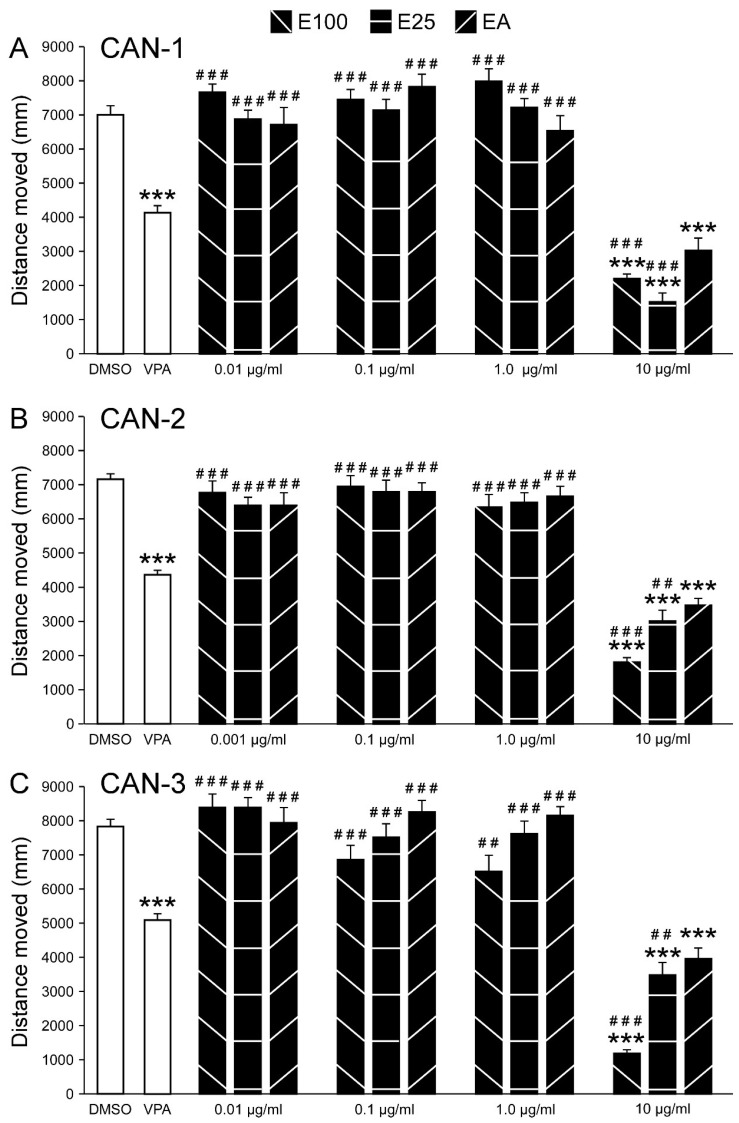
Effects of CAN-1, CAN-2, and CAN-3 extracts on PTZ-induced larval movement. Total distance moved (mm) by zebrafish larvae during 30 min of PTZ exposure following pre-treatment with DMSO; VPA (positive control); or varying concentrations of extracts from CAN-1 (**A**), CAN-2 (**B**), and CAN-3 (**C**) obtained through different extraction methods (E100, E25, and EA). *** *p* < 0.001 vs. PTZ; vs. VPA; ^##^ *p* < 0.01 vs. VPA; ^###^ *p* < 0.001 vs. VPA.

**Figure 4 biomolecules-15-00654-f004:**
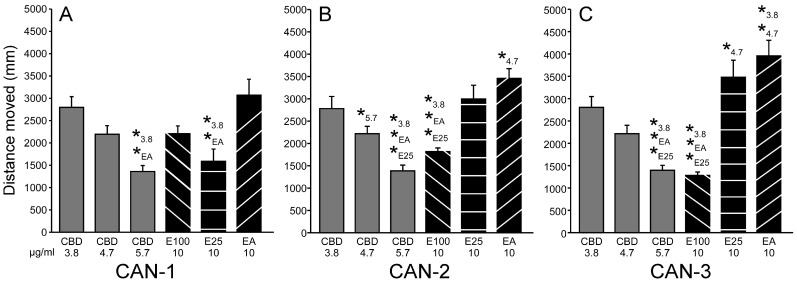
Effective CBD and extract concentrations. Comparison of effects of CAN-1, CAN-2, CAN-3, and CBD on PTZ-induced larval movement. Total distance moved (mm) by zebrafish larvae during 30 min of PTZ exposure following pre-treatment with CBD (n = 48/group) or 10 µg/mL extracts from cannabis strains CAN-1 (**A**), CAN-2 (**B**), and CAN-3 (**C**) obtained using different extraction methods (E100, E25, and EA; n = 36/group). * represents a significant difference compared to indicated test solution.

**Table 1 biomolecules-15-00654-t001:** The cannabinoid contents (µg/mL) in the 10 µg/mL solutions of the different cannabis extracts.

Strain	CAN-1			CAN-2			CAN-3		
**Extract**	**E25**	**E100**	**EA**	**E25**	**E100**	**EA**	**E25**	**E100**	**EA**
**CBDA**	0.032	0.017	0.026	4.545	1.557	4.791	4.055	2.199	4.164
**CBD**	0.015	0.024	0.012	0.763	5.174	0.884	0.201	2.661	0.215
**THCA**	5.722	0.740	4.571	0.259	0.133	0.274	2.135	0.491	2.166
**Δ^9^-THC**	0.816	5.149	0.604	0.097	0.852	0.101	0.299	2.343	0.291
**CBGA**	0.417	0.124	0.304	0.083	0.040	0.092	0.086	0.057	0.087
**CBG**	0.045	0.169	0.035	0.033	0.117	0.036	0.041	0.079	0.042
**CBCA**	0.438	0.120	0.382	0.359	0.077	0.378	0.236	0.106	0.247
**CBC**	0.033	0.230	0.025	0.063	0.302	0.069	0.033	0.149	0.035
**CBNA**	0.112	0.024	0.083	0.010	0.010	0.011	0.066	0.029	0.058
**CBN**	0.027	0.070	0.022	0.016	0.028	0.017	0.029	0.066	0.029
**CBDV**	0	0	0	0.018	0.028	0.018	0.026	0.037	0.026
**THCV**	0.020	0.071	0.017	0	0.016	0	0.022	0.039	0.022

**Table 2 biomolecules-15-00654-t002:** The terpene contents (µg/mL) in the 10 µg/mL solutions of the different cannabis extracts.

Strain	CAN-1			CAN-2			CAN-3		
**Extract**	**E25**	**E100**	**EA**	**E25**	**E100**	**EA**	**E25**	**E100**	**EA**
**Alpha Bisabolol**	0.0063	0.0076	0.0068	0.0151	0.0126	0.0140	-	-	-
**Alpha Cedrene**	0.0087	0.0024	0.0090	0.0036	0.0007	0.0042	0.0017	0.0011	0.0035
**Alpha Humulene**	0.0044	0.0017	0.0045	0.0018	0.0007	0.0019	0.0011	0.0009	0.0020
**Alpha Pinene**	-	-	-	-	-	0.0008	-	-	0.0010
**Alpha Terpineol**	0.0012	0.0005	0.0012	0.0010	-	0.0010	0.0010	-	0.0018
**Beta Caryophyllene**	0.0109	0.0030	0.0113	0.0045	0.0008	0.0052	0.0021	0.0013	0.0044
**Beta Eudesmol**	0.0039	0.0042	0.0045	0.0047	0.0040	0.0044	0.0012	0.0030	0.0046
**Beta Pinene**	0.0001	0.00007	0.0003	-	0.00007	0.0004	0.0002	-	0.0005
**Borneol**	0.0019	0.0024	0.0022	0.0027	0.0027	0.0045	0.0026	0.0050	0.0054
**Cedrol**	0.0009	0.0008	0.0010	0.0008	0.0007	0.0008	0.0007	0.0012	0.0013
**Fenchol**	0.0010	0.0004	0.0010	0.0008	-	0.0008	0.0008	-	0.0016
**(-) Guaiol**	0.0034	0.0040	0.0036	0.0047	0.0035	0.0044	0.0020	0.0030	0.0039
**Limonene (2)**	0.0005	0.0002	0.0008	-	-	0.0002	-	-	0.0005
**Myrcene**	0.0012	0.0005	0.0033	0.0006	-	0.0024	0.0007	-	0.0022
**Terpinolene**	0.0006	0.0003	0.0006	0.0006	-	0.0006	0.0006	-	0.0010
**Trans Nerolidol**	-	-	-	0.0021	0.0018	0.0021	0.0008	-	-

## Data Availability

The original data presented in this study are openly available in the Appendix A.

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
