# Peer review of "The Anticonvulsant Effects of Different Cannabis Extracts in a Zebrafish Model of Epilepsy"

_biomolecules, 2025, doi:10.3390/biom15050654_

Round 1

Reviewer 1 Report (Previous Reviewer 1)

Comments and Suggestions for Authors

I agree with changes made in reply to my review, the paper could be prointed

Author Response

Dear Reviewer,

Thank you very much for your kind approval of our work.

We appreciate it very much.

BEST REGARDS,

The authors

Reviewer 2 Report (Previous Reviewer 2)

Comments and Suggestions for Authors

As stated in the previous report, the paper is well-written, the data are new and interesting, the experiments are properly designed, and the conclusions are fair. I confirm that the authors chose the proper bibliography to support their data. The revised manuscript is also improved compared to the previous version. New statements, descriptions, figures, and further discussion definitely enhanced the quality of the manuscript, thus, I confirm my positive feedback for the suitability of this paper for publication on biomolecules.

Author Response

Dear Reviewer,

Thank you very much for your kind approval of our work.

We appreciate it very much.

WE WISH YOU ALL THE BEST,

The authors

Reviewer 3 Report (Previous Reviewer 3)

Comments and Suggestions for Authors

The authors have addressed most of the reviewers' concerns in the revised manuscript. However, some minor details remain missing and need to be addressed.

1)  With the addition of Figure 1, the reviewer has a much better understanding of the experimental design. However, the authors should include a column for the viability test in Figure 1. The authors should also mention if after the viability test, zebrafish were excluded after the viability test (and how many).  

2) In the supplemental table S7 (CAN-2), the mean dark-light for E100-10mg is a positive value (around 6), suggesting more locomotion in the light phase, whereas all other mean dark-light values in S5 and S7 tables are negative (Range: -20 to -80). This can potentially be an outlier. The authors should conduct an appropriate test to determine if CAN-2 E100-10 is indeed an outlier and perhaps exclude it from data analysis. 

Author Response

Dear Reviewer,
We would like to sincerely thank you for your thorough review of our manuscript. Your insightful comments prompted us to conduct an additional and comprehensive examination of all the results presented in the manuscript.

We wish you all the best,

The Authors.

Point by point answers:

Comment 1: With the addition of Figure 1, the reviewer has a much better understanding of the experimental design. However, the authors should include a column for the viability test in Figure 1. The authors should also mention if after the viability test, zebrafish were excluded after the viability test (and how many).  

Response 1: We have carefully considered and addressed these concerns by modifying the title of column number 5 in Fig 1. instead of Dark (3 min), we modified it to: Viability test (Dark; 3min). We also added a sentence to this paragraph:

Lines 222: None of the larvae were excluded from the experiments after the viability test.

Comment 2: In the supplemental table S7 (CAN-2), the mean dark-light for E100-10mg is a positive value (around 6), suggesting more locomotion in the light phase, whereas all other mean dark-light values in S5 and S7 tables are negative (Range: -20 to -80). This can potentially be an outlier. The authors should conduct an appropriate test to determine if CAN-2 E100-10 is indeed an outlier and perhaps exclude it from data analysis.

Response 2: The viability test assessed the larvae’s responsiveness to sudden darkness exposure, a condition that typically induces an increase in locomotor activity. The test was conducted following a 30-minute treatment with the tested compounds. Results indicated that all groups, with the exception of one, exhibited the expected increase in movement in response to darkness. Prior to treatment (treatment condition), all groups demonstrated similar behavioural responses. The lack of response to darkness observed in the exceptional group may be attributed to factors other than sedation. In addition, we observed that the E100 group from CAN 3 responded to darkness exposure and exhibited a slightly better, although not statistically significant, reduction in PTZ-induced hyperactivity compared to the E100 group from CAN 2. Had the E100 group from CAN 2 induced sedation in the larvae, this effect would likely have been reflected in the subsequent PTZ experiment; however, no indication of sedation was observed under these conditions.

We also added these 2 paragraphs in the Results section:

Lines 215-222: Furthermore, we conducted a viability test, which revealed an increase in movement following exposure to darkness (3 minutes), suggesting that the cannabis extracts did not affect the behavioral reaction to darkness (Tables S4–S9). An exception was ob-served in one group (Table S7; CAN 2 – E100 – 10 µg/ml), which did not show an in-crease in movement following darkness exposure. However, the behavioral results in Table S1 did not reveal any abnormal total distance moved for this treatment (E100 – 10 µg/ml; CAN 1, 2 and 3). None of the larvae were excluded from the experiments after the viability test.

Lines 267-272: E100 from CAN 3 (10 µg/ml) reduced PTZ-induced hyperactivity slightly better than E100 from CAN 2 (10 µg/ml). This was observed despite the viability test results, which showed that E100 from CAN 3 (10 µg/ml) led to an increase in movement following darkness exposure, whereas E100 from CAN 2 (10 µg/ml) did not (Tables S7 and S9), suggesting that E100 from CAN 2 (10 µg/ml) exerted other behavioural effects rather than sedation.

In Supplementary material:

Page 7: Viability Test (Dark; 3 min)

This manuscript is a resubmission of an earlier submission. The following is a list of the peer review reports and author responses from that submission.

Round 1

Reviewer 1 Report

Comments and Suggestions for Authors

I suggest to discuss in more details the use of these substancesin other seizure models

Author Response

Reviewer1  
Comment 1: The paper presented contains the detailed description of the effects of cannabis derived substances stressing their benefits in comparison to controls and VPA effects. There are no negative comments concerning the whole work and the results presented. 

Response 1: We sincerely appreciate your positive feedback on our manuscript. Your acknowledgment of our work encourages us to continue our research in this area.

Comment 2: Although the use of fish larvae as a single model has certain limitations which needs to be stressed in the text (either in Introduction or in Discussion sections). In other words more detailed description of the behavioral signs of seizures in fish larvae is needed in comparison to more common seizure-like activity in rodent models. It means - why the distance travelled by a larvae is the index of seizure propensity?

Response 2: We appreciate the reviewer's thoughtful evaluation of our methodology. While we acknowledge that zebrafish exhibit multiple behavioral manifestations during PTZ-induced seizures, substantial evidence supports the use of distance traveled as a reliable and sufficient proxy for seizure activity in larval zebrafish.  

Distance traveled (or integrated locomotor activity) remains a sensitive and comprehensive indicator of seizure severity, as this metric inherently captures both the frequency and magnitude of abnormal movements, increasing proportionally with more frequent or vigorous convulsive bouts. Studies assessing multiple parameters often find that distance or velocity robustly characterizes the seizure phenotype without requiring complex scoring. In fact, a systematic review of chemically induced seizure models in zebrafish (PMID: 37801749) confirms that distance traveled is among the most commonly assessed behavioral outcomes, serving as a primary endpoint in nearly 50% of the 178 studies reviewed that included behavioral analyses. In addition, studies have demonstrated that antiepileptic drugs reduce locomotor activity in a dose-dependent manner, further validating distance traveled as a reliable endpoint (e.g., PMID: 38542281, 32186159). Finally, locomotor activity, quantified by distance traveled, strongly correlates with seizure severity in zebrafish models. Hence, while no single behavioral metric is universally superior, distance traveled is widely favored for its simplicity, quantitative nature, and ability to reflect the cumulative locomotor hyperactivity characteristic of the seizure-like state.

We shortly refer to these issues in the 3rd paragraph of the Introduction (Lines 79-91), as well as to the limitation section of the Discussion (Lines 404-8), and would be happy to clarify this point further in the manuscript if needed.

Comment 3: The data obtained are discussed solely in connection with clinical data and there are no mentions concerning the epilepsy signs in other preclinical model. It means that some details in model description are warranted.

Response 3: We believe that discussing epilepsy signs in other preclinical models falls outside the scope of our manuscript, which focuses specifically on zebrafish larvae. However, we have added a brief sentence in the 4th paragraph of the Introduction (Lines 81-2), with additional data in the limitation section of the Discussion (Lines 404-8), and provided a reference to more comprehensive reviews for interested readers (e.g., PMID: 36461665 and 35111370). That said, if the reviewer considers this information essential, we would be happy to incorporate it into the revised manuscript.

Reviewer 2 Report

Comments and Suggestions for Authors

Jackson et al. – biomolecules:

The authors evaluated the anticonvulsant effects of several cannabis extracts in a zebrafish model of epilepsy. They took advantage of the pentylenetetrazole (PTZ)-induced hyperactivity model in zebrafish, and they characterized the efficacy of different cannabis strains. Moreover, they compared the effectiveness of these compounds with the valproic acid (VPA) and the pure cannabidiol (CBD). The authors showed that different extracts significantly reduced movement compared to PTZ and VPA. Thus, reinforcing previous evidence supporting the clinical potential of both CBD and whole cannabis extracts for seizure control. The paper is well-written, the data are new and interesting, the experiments are properly designed, and the conclusions are fair. The authors chose the proper bibliography to support their data—reasons why the paper is suitable for publication on biomolecules.

Minor comments:

Results

For consistency, authors might consider choosing a similar range for the Y-axis among figures (something like 11000 mm).

Discussion

Pg. 10, lines 338-354; authors should consider rephrasing this paragraph since it seems like a list of different concentrations in different papers.

Author Response

Reviewer 2
Comment 1: The authors evaluated the anticonvulsant effects of several cannabis extracts in a zebrafish model of epilepsy. They took advantage of the pentylenetetrazole (PTZ)-induced hyperactivity model in zebrafish, and they characterized the efficacy of different cannabis strains. Moreover, they compared the effectiveness of these compounds with the valproic acid (VPA) and the pure cannabidiol (CBD). The authors showed that different extracts significantly reduced movement compared to PTZ and VPA. Thus, reinforcing previous evidence supporting the clinical potential of both CBD and whole cannabis extracts for seizure control. The paper is well-written, the data are new and interesting, the experiments are properly designed, and the conclusions are fair. The authors chose the proper bibliography to support their data—reasons why the paper is suitable for publication on biomolecules.
Response 1: We sincerely appreciate the reviewer’s thoughtful and kind words.

Minor comments:
Results
Comment 2: For consistency, authors might consider choosing a similar range for the Y-axis among figures (something like 11000 mm).
Response 2: In accordance with your request, the Y-axis of the first two result figures (Figures 2 and 3 in the revised manuscript) was set to a maximum value of 9000 mm (please note reduced average scores for the CBD groups following increased sample sizes). As for Figure 4 in the revised manuscript, we believe that using the same scale is less appropriate, as it makes the differences between groups harder to discern (please see below). Therefore, the figure in the manuscript has been formatted with a maximum value of 5000 mm. However, if the reviewer still believes that presenting the figure below would be more beneficial for the reader, we are willing to adjust the manuscript accordingly.

Discussion

Comment 3: Pg. 10, lines 338-354; authors should consider rephrasing this paragraph since it seems like a list of different concentrations in different papers.
Response 3: We thank the reviewer for this comment and made corrections accordingly in the revised manuscript, as requested (Lines 388-393). It is now stated that: “Our results suggest a beneficial effect of pure CBD at a relatively higher concentration of 5.7 µg/ml (or 18 µM) compared to prior studies [41–43], which reported significant effects at concentrations of 0.6–4 µM. This may be explained by protocol differences, including variations in exposure time, PTZ concentration, and monitoring conditions, which influenced the effective CBD dose. These differences highlight the need to verify optimal CBD dosing within each study’s specific conditions before making comparisons.”

Reviewer 3 Report

Comments and Suggestions for Authors

This is an interesting study investigating the anticonvulsant effects of pure cannabidiol (CBD) and cannabis extracts, using the pentylenetetrazole (PTZ) model of seizure-like activity in zebrafish. Although the results are interesting, the manuscript falls short in several aspects (particularly in experimental design) described in detail below:

  1. Authors use distance moved as a sole proxy (behavioral correlate) of seizures in larval zebrafish. However, PTZ zebrafish models exhibit distinct behaviors reminiscent of clinical seizures. Prior studies score seizures in zebrafish by characterizing parameters like 1) short swim, 2) increased swimming activity and high frequency of opercular movement, 3) erratic movements, (4) circular movements, (5) clonic seizure-like behavior, (6) fall to the bottom of the tank and tonic seizure-like behavior, (7) death (PMID: 23349914, 22542883). Therefore, relying on distance traveled or locomotion alone can lead to misinterpretation of the data (i.e. sedation caused by CBD or cannabis extracts).
  2. Most studies with zebrafish as disease models studying behavior utilize adult fish (PMID: 22542883, 28824436). Larval zebrafish have a distinct advantage over adults, in fluorescent imaging (single and multiphoton) of CNS-wide activity associated with behaviors, due to their small size and transparency. However, the authors do not utilize fluorescent imaging, nor do they investigate the developmental effects of early-life seizures. Therefore, a justification for using larval zebrafish over adults is needed.  
  3. Key controls from the experimental design are missing. I would recommend the authors add CBD/extracts (0.01mg/mL) with no PTZ for 30 mins and no treatment with no PTZ controls. These controls will help establish whether CBD and cannabis extracts have sedative effects on larvae that hinder locomotion.
  4. The authors mentioned that the locomotion assay was performed using a lightproof recording chamber equipped with an infrared camera. This should allow authors to perform behavioral assays in the dark phase using IR illumination, where zebrafish locomotion is significantly higher. Yet, according to my understanding, the authors performed the experiments under light. They then tested the increased larval locomotion in the dark phase only during the post-hoc viability test, to test if suppression of locomotion was due to anti-seizure effects and not sedation. This suggests a low signal-to-noise ratio for locomotion measurements. The authors should provide a detailed rationale for this choice in experimental design.
  5. It is not clear whether the larval zebrafish were subjected to PTZ during the viability test?
  6. A comparison of locomotion data from the viability test (Δ locomotion: dark-light) between individual groups will be helpful for interpretation and should be reported.
  7. Why did the authors choose to have no technical replicates for CBD positive control, unlike other groups? (at least 3 replications are preferred)
  8. The current experimental design with drug preincubation followed by seizure induction, only tests the efficacy of CBD and cannabis extracts as preventative agents. Since a lot of AEDs are rescue medications, it'll be interesting to see if CBD extract arrests seizures/reduces seizure severity, when administered after PDZ application. alternatively, this limitation of the study should be discussed in the manuscript.

Author Response

Reviewer 3
Comment 1: This is an interesting study investigating the anticonvulsant effects of pure cannabidiol (CBD) and cannabis extracts, using the pentylenetetrazole (PTZ) model of seizure-like activity in zebrafish. Although the results are interesting, the manuscript falls short in several aspects (particularly in experimental design) described in detail below.
Response 1: We sincerely thank the reviewer for their comments and for highlighting the constructive issues below. We have carefully considered and addressed these concerns and hope that our revisions adequately resolve them.

Major comments:
Comment 2: Authors use distance moved as a sole proxy (behavioral correlate) of seizures in larval zebrafish. However, PTZ zebrafish models exhibit distinct behaviors reminiscent of clinical seizures. Prior studies score seizures in zebrafish by characterizing parameters like 1) short swim, 2) increased swimming activity and high frequency of opercular movement, 3) erratic movements, (4) circular movements, (5) clonic seizure-like behavior, (6) fall to the bottom of the tank and tonic seizure-like behavior, (7) death (PMID: 23349914, 22542883). Therefore, relying on distance traveled or locomotion alone can lead to misinterpretation of the data (i.e. sedation caused by CBD or cannabis extracts).
Response 2: We agree that PTZ-induced seizures in adult zebrafish encompass multiple behavioral phenotypes beyond locomotion alone, as highlighted in the studies suggested by the reviewer (PMID: 23349914, 22542883). However, distance traveled remains a well-established and reliable parameter for assessing seizure activity in zebrafish larvae. Numerous studies have shown that PTZ exposure significantly increases locomotion, with strong correlations to seizure severity.  While behaviors such as erratic movements, circular swimming, and opercular activity are indeed observed during seizures, distance moved provides an objective and quantifiable measure that has been extensively validated in larval zebrafish seizure models (e.g., PMID: 32585475, 30949046, 24291671, 15730879, 22730455). The use of automated tracking systems further ensures precise measurement of hyperlocomotion, reducing observer bias and enhancing reproducibility. 

We are now addressing this issue in the 3th paragraph of the Introduction (Lines 79-91), with additional data in the limitation section of the Discussion (Lines 404-8). We appreciate the reviewer’s insights and would be happy to clarify this point further in the manuscript if needed.

With regard to the concern of possible sedation, we are now presenting data that verify the lack of sedative effects caused by the cannabis extracts (Lines 237-41 and Tables S1-9).

Comment 3: Most studies with zebrafish as disease models studying behavior utilize adult fish (PMID: 22542883, 28824436). Larval zebrafish have a distinct advantage over adults, in fluorescent imaging (single and multiphoton) of CNS-wide activity associated with behaviors, due to their small size and transparency. However, the authors do not utilize fluorescent imaging, nor do they investigate the developmental effects of early-life seizures. Therefore, a justification for using larval zebrafish over adults is needed.  
Response 3: Thank you for your comment. While it is true that many behavioral studies using zebrafish as disease models are conducted in adults, larval zebrafish offer distinct advantages in seizure research, even beyond their compatibility with fluorescent imaging. Larval zebrafish are widely preferred for studying PTZ-induced convulsions due to their practical advantages and suitability for high-throughput behavioral screening. Although both larval and adult zebrafish exhibit clear seizure phenotypes following convulsant exposure, several key factors make larvae the optimal choice for such studies:

I.    High-Throughput Screening Potential: Unlike adults, larval zebrafish can be studied in multi-well plate formats, such as 96-well plates, significantly increasing the scalability of experiments. This setup enables parallel testing of multiple conditions, facilitating large-scale drug screening related to epilepsy and convulsions. The ability to assess large numbers of larvae in a controlled environment improves the statistical power and reproducibility of findings (also see PMID: 35743088).
II.    Simplified behavioral analysis: While adult zebrafish exhibit more complex behaviors, the simplified behavioral repertoire of larvae is sufficient and appropriate for our specific research questions. The simpler behavioral repertoire of larvae allows for more straightforward quantification of seizure-like activities. Baraban et al. (2005) demonstrated that locomotor activity in larvae could be effectively used to study PTZ-induced seizures (PMID: 15730879).
III.    Correlation with electrophysiology: Changes in locomotor activity in larval zebrafish have been shown to correlate well with electrophysiological measures of seizure activity. Afrikanova et al., demonstrated a strong correlation between locomotor activity and electrical discharges in the brain of PTZ-induced seizures in zebrafish larvae (PMID: 23342097).
IV.    Established protocols: There is a wealth of established protocols and literature supporting the use of larval zebrafish in epilepsy research, particularly for drug screening purposes increases in larval movement, indicating the potential utility of zebrafish as a high-throughput in vivo model for AED discovery (PMID: 37801749 and 37701853). 
V.    Sensitivity to pharmacological interventions: Our measure of distance moved in larvae has proven sensitive enough to detect the effects of various compounds, including CBD and cannabis extracts. This sensitivity is crucial for our research objectives.
VI.    3Rs Principles: Employing zebrafish larvae aligns with the principles of Replacement, Reduction, and Refinement (3Rs), minimizing the use of adult animals in biomedical research. 
While we acknowledge that fluorescent imaging and developmental studies of early-life seizures are valuable approaches, our study primarily focuses on behavioral seizure assessment and pharmacological testing, where larval zebrafish remain a highly suitable model. 
We appreciate the opportunity to clarify this choice and can further elaborate on this rationale in the manuscript if necessary. As for now, we shortly refer to this issue in the 3rd paragraph of the Introduction (Lines 79-80) and provided a reference to a more comprehensive review for interested readers.

Comment 4: Key controls from the experimental design are missing. I would recommend the authors add CBD/extracts (0.01mg/mL) with no PTZ for 30 mins and no treatment with no PTZ controls. These controls will help establish whether CBD and cannabis extracts have sedative effects on larvae that hinder locomotion.
Response 4: We thank the reviewer for this comment and added the requested control data (formerly donated as ‘data not shown’) for all doses of the extracts to the revised manuscript in the Result section (1.1.1 Lack of sedative effects; Lines 238-41) and in Tables S1-3. In these tables, larval movement during the treatment phase (no PTZ) are presented, and the results indeed demonstrate that the extracts do not cause sedative effects. Regarding the control of no treatment and no PTZ, this data is available for all analyses (please see DMSO data in all Tables and the new Figure 1 for visualization of the procedure). Although we do not have similar data for CBD, former publications have already established a lack of sedative effects for CBD in similar concentrations (e.g., PMID: 37759798).

Comment 5: The authors mentioned that the locomotion assay was performed using a lightproof recording chamber equipped with an infrared camera. This should allow authors to perform behavioral assays in the dark phase using IR illumination, where zebrafish locomotion is significantly higher. Yet, according to my understanding, the authors performed the experiments under light. They then tested the increased larval locomotion in the dark phase only during the post-hoc viability test, to test if suppression of locomotion was due to anti-seizure effects and not sedation. This suggests a low signal-to-noise ratio for locomotion measurements. The authors should provide a detailed rationale for this choice in experimental design.
Response 5: Thank you for your insightful comment. We acknowledge the importance of lighting conditions in zebrafish behavioral assays, as locomotion in larval zebrafish is significantly influenced by light-dark transitions. We have now added the data regarding exposure to dark (Tables S4-9), which was formerly donated as ‘data not shown’, and relate to it in the Results (1.1.1 Lack of sedative effects; Lines 238-41)

Zebrafish larvae at 5 days post-fertilization (dpf) exhibit fully functional visual and locomotor systems, enabling robust behavioral responses to light stimuli. It is known that the larval zebrafish has a preference for light and an aversion for dark (e.g., PMID: 18952124 and 20540966), a tendency that is reversed in adult zebrafish. Hence, when using convulsion assays in zebrafish larvae, controlled light conditions offer several advantages over darkness, enhancing the reliability and depth of experimental outcomes. For example, exposure to light significantly increases the locomotor activity in zebrafish larvae under PTZ, providing a 2–3 times higher dynamic range in movement. This heightened activity facilitates more sensitive and accurate detection of seizure events (PMID: 3750608). 
In our study, larvae were raised under 14:10 hour light-dark cycle and experiments were performed during the day. Maintaining light conditions aligns with the natural circadian rhythms of zebrafish, which are diurnal animals, and hence reduces stress-induced artifacts and minimizes variability between experiments. By providing a consistent light-dark cycle, we ensured that experiments are conducted at optimal times, reflecting the natural activity patterns of zebrafish and thereby reducing inter-experimental variability. Indeed, several former studies using zebrafish larvae for convulsions were performed under light conditions (e.g., PMID: 37506089 and 30949046).
In summary, controlled light exposure in zebrafish convulsion assays enhances behavioral responses, aligns with natural circadian rhythms, thereby improving the reliability and interpretability of the experiments. 

Comment 6: It is not clear whether the larval zebrafish were subjected to PTZ during the viability test?
Response 6: We have added a figure to visualize the experimental procedure and to avoid such confusion (Figure 1 in the revised manuscript). Following the treatment phase, viability was conducted visually and by placing the larvae in darkness for 3 minutes in E3 medium to observe their movement patterns, as is now clarified in the appropriate Method section (2.4 Seizure Assay; Lines 172-5). 

Comment 7: A comparison of locomotion data from the viability test (Δ locomotion: dark-light) between individual groups will be helpful for interpretation and should be reported.
Response 7: We thank the reviewer for this comment and added the requested control data to the revised manuscript, as noted above (Response 5). Larvae that completely lost mobility were excluded from the experiment, while a significant increase in movement during the dark phase further suggested that the cannabis extracts did not function as sedatives (Tables S4–9)

Comment 8: Why did the authors choose to have no technical replicates for CBD positive control, unlike other groups? (at least 3 replications are preferred)
Response 8: We initially conducted four replications for this condition; however, the data was unavailable due to human error. In response to the reviewer's comment, we reinvestigated and discovered the data in a mislabeled folder. This data is now included in the revised manuscript, with n = 48 per group. It should be noted that the increased sample size did not affect the results or conclusions. 

Comment 9: The current experimental design with drug preincubation followed by seizure induction, only tests the efficacy of CBD and cannabis extracts as preventative agents. Since a lot of AEDs are rescue medications, it'll be interesting to see if CBD extract arrests seizures/reduces seizure severity, when administered after PDZ application. alternatively, this limitation of the study should be discussed in the manuscript.
Response 9: The reviewer is correct, and this issue is now addressed in the limitation section of the revised manuscript (Lines 414-7).